# A High-Performance and Durable Direct-Ammonia Symmetrical Solid Oxide Fuel Cell with Nano La_0.6_Sr_0.4_Fe_0.7_Ni_0.2_Mo_0.1_O_3−δ_-Decorated Doped Ceria Electrode

**DOI:** 10.3390/nano14080673

**Published:** 2024-04-12

**Authors:** Hao Jiang, Zhixian Liang, Hao Qiu, Yongning Yi, Shanshan Jiang, Jiahuan Xu, Wei Wang, Chao Su, Tao Yang

**Affiliations:** 1School of Energy and Power, Jiangsu University of Science and Technology, Zhenjiang 212100, China; jianghao@stu.just.edu.cn (H.J.); liangzhixian@stu.just.edu.cn (Z.L.); qiuhao@stu.just.edu.cn (H.Q.); jss522@just.edu.cn (S.J.); 2State Key Laboratory of Materials-Oriented Chemical Engineering, College of Chemical Engineering, Nanjing Tech University, Nanjing 211816, China; 202262104022@njtech.edu.cn (Y.Y.); wangwei@njtech.edu.cn (W.W.); 3School of Science, Jiangsu University of Science and Technology, Zhenjiang 212100, China; jiahuanxu@just.edu.cn; 4Future Technology School, Shenzhen Technology University, Shenzhen 518118, China

**Keywords:** ammonia, symmetrical solid oxide fuel cells, nanoparticles, perovskite oxide, molybdenum doping

## Abstract

Solid oxide fuel cells (SOFCs) offer a significant advantage over other fuel cells in terms of flexibility in the choice of fuel. Ammonia stands out as an excellent fuel choice for SOFCs due to its easy transportation and storage, carbon-free nature and mature synthesis technology. For direct-ammonia SOFCs (DA-SOFCs), the development of anode catalysts that have efficient catalytic activity for both NH_3_ decomposition and H_2_ oxidation reactions is of great significance. Herein, we develop a Mo-doped La_0.6_Sr_0.4_Fe_0.8_Ni_0.2_O_3−δ_ (La_0.6_Sr_0.4_Fe_0.7_Ni_0.2_Mo_0.1_O_3−δ_, LSFNM) material, and explore its potential as a symmetrical electrode for DA-SOFCs. After reduction, the main cubic perovskite phase of LSFNM remained unchanged, but some FeNi_3_ alloy nanoparticles and a small amount of SrLaFeO_4_ oxide phase were generated. Such reduced LSFNM exhibits excellent catalytic activity for ammonia decomposition due to the presence of FeNi_3_ alloy nanoparticles, ensuring that it can be used as an anode for DA-SOFCs. In addition, LSFNM shows high oxygen reduction reactivity, indicating that it can also be a cathode for DA-SOFCs. Consequently, a direct-ammonia symmetrical SOFC (DA-SSOFC) with the LSFNM-infiltrated doped ceria (LSFNM-SDCi) electrode delivers a superior peak power density (PPD) of 487 mW cm^−2^ at 800 °C when NH_3_ fuel is utilised. More importantly, because Mo doping greatly enhances the reduction stability of the material, the DA-SSOFC with the LSFN-MSDCi electrode exhibits strong operational stability without significant degradation for over 400 h at 700 °C.

## 1. Introduction

For the protection of the environment and sustainable development, it is necessary to embrace green energy and to develop new power generation methods and technologies [1,2,3]. The solid oxide fuel cell (SOFC) is a novel electrochemical conversion device that can immediately transform chemical energy into electrical energy, thereby enhancing the efficiency of conversion and minimising pollutants [4,5]. More importantly, SOFCs stand out from other forms of fuel cell due to their notable fuel adaptability. Any combustible chemical can be used as the fuel of an SOFC, including hydrogen (gas), ethanol (liquid) and carbon (solid) [6,7]. Hydrogen (H_2_) is currently the most frequently utilised fuel in SOFCs. However, its storage and transport pose considerable difficulties due to its extremely low liquefaction temperature [8], greatly hindering the commercial application of H_2_. Carbon-free ammonia (NH_3_), the best carrier for H_2_, is an attractive alternative fuel with several advantages [9,10]. For example, NH_3_ exhibits higher energy density and enhanced safety due to the ease of its detection [11]. Furthermore, the relatively higher liquefaction temperature of ammonia simplifies its storage and transportation [12]. Therefore, the use of green NH_3_ as an energy source for SOFCs to generate electricity can lead to a sustainable, clean energy system.

Presently, Ni-based cermet is the prevailing choice for anode materials in SOFCs [13]. However, for the DA-SOFC, under high NH_3_ concentration, Ni is susceptible to significant coarsening. This causes the anode microstructure to deteriorate and results in a rapid decline in cell performance [14]. Therefore, the development of anode materials suitable for operation in an ammonia atmosphere is the focus of current research on DA-SOFCs. Perovskite oxides possess superb electrocatalytic activity and can be a good alternative to nickel-based cermet anodes. Recently, many perovskite oxide anodes have been developed for DA-SOFCs. Some examples are Pr_0.6_Sr_0.4_Co_0.2_Fe_0.75_Ru_0.05_O_3−δ_, La_0.45_Sr_0.45_Ti_0.9_Ni_0.1_O_3−δ_ and La_0.52_Sr_0.28_Ti_0.94_Ni_0.03_Co_0.03_O_3−δ_ [15,16,17]. However, so far, there are still too few low-cost and high-performance perovskite anode materials for DA-SOFCs, and more such anodes need to be developed. In addition, as we all know, perovskite oxides make superb cathode materials for SOFCs. If a perovskite oxide simultaneously possesses good activity for oxygen reduction reaction (ORR), ammonia decomposition reaction (NDR) and hydrogen oxidation reaction (HOR), then symmetrical SOFC (SSOFC) technology can be applied to the development of a DA-SSOFC. Recently, there has been an increasing interest in SSOFCs that utilise a single material for both the cathode and anode due to the potential cost reduction and simplified manufacturing process, which can accelerate the commercialisation of SOFCs [18,19]. La_0.6_Sr_0.4_Fe_x_Ni_1−x_O_3−δ_ (LSFNx) is a typical perovskite oxide electrode for SSOFCs [20]. Due to the easy exsolution of Ni in the LSFNx under a reducing atmosphere, the reduced LSFNx shows excellent catalytic activity. At the same time, LSFNx also presents good ORR activity. In addition to being used as an electrode material for SSOFCs, LSFNx has also been developed as an electrode for the symmetrical solid oxide electrolysis cell (SSOEC) and reversible solid oxide cells (RSOC) [21,22]. However, LSFNx has a big problem, which is its poor stability in reducing atmospheres at high temperatures (above 750 °C) [23].

Doping is a highly effective method for enhancing the structural stability and electrocatalytic activity of catalysts [24]. The incorporation of dopants with elevated oxidation states, such as molybdenum (Mo), causes the B-site element to decrease from the high to the low valence state. When exposed to a reducing environment, the B-site elements are selectively reduced to metal atoms, while the dopant aids in preserving the structure. For example, Hou et al. reported that the structural stability of La_0.6_Sr_0.4_Fe_0.9_Ni_0.1_O_3−δ_ perovskite oxide within a reducing environment is greatly enhanced by doping it with high-valent Mo [25]. Li et al. demonstrated that an SOEC with a hollow La_0.6_Sr_0.4_Ni_0.2_Fe_0.75_Mo_0.05_O_3−δ_ electrode shows good activity for CO_2_ electrolysis [26]. Lu et al. found that doping Mo into Pr_0.6_Sr_0.4_Fe_0.8_Ni_0.2_O_3−δ_ boosted the hydrogen oxidation reaction (HOR) performance [27]. The research conducted by Xiao et al. has demonstrated that Mo doping can enhance SrFeO_3−δ_-based perovskites’ resistance to sintering [28].

Herein, to improve the stability of LSFNx, we developed a Mo-doped La_0.6_Sr_0.4_Fe_0.8_Ni_0.2_O_3−δ_ (La_0.6_Sr_0.4_Fe_0.7_Ni_0.2_Mo_0.1_O_3−δ_, LSFNM) catalyst and used it as an electrode for DA-SSOFCs for the first time. We found that its reduction stability was significantly improved by the doping. A remarkable catalytic activity for NDR makes LSFNM a highly efficient anode material for DA-SSOFCs. Furthermore, LSFNM also displayed good activity for ORR. The SSOFC with the LSFNM-infiltrated Sm_0.2_Ce_0.8_O_1.9_ (LSFNM-SDCi) nanocomposite electrode achieved a high value for peak power density (PPD) of 487 mW cm^−2^ at 800 °C in NH_3_. Most significantly, the DA-SSOFC demonstrated stable operation when fuelled with both hydrogen and ammonia, maintaining performance without any significant degradation for over 400 h at 700 °C.

## 2. Materials and Methods

### 2.1. Materials Synthesis

The LSFNM and LSFN powders were synthesised via a complexing sol-gel method. Taking LSFNM as an example, La(NO_3_)_3_·6H_2_O, Sr(NO_3_)_2_, Fe(NO_3_)_3_·9H_2_O, Ni(NO_3_)_3_·6H_2_O and (NH_4_)_6_Mo_7_O_24_·4H_2_O were dissolved in deionised water in stoichiometric amounts. The ethylenediaminetetraacetic acid (EDTA) and citric acid were added in sequence. The mole ratio of EDTA to total metal ions to citric acid was 1:1:2. The pH value of the solution was adjusted to 7–8 with ammonia solution. When the solution became gel-like, it was dried at 250 °C for 5 h, and then calcined at 1000 °C for 5 h to gain the LSFNM powders. The SDC used in this study were purchased from Qingdao Terio Corporation, Qingdao, China. The LSFNM or LSFN precursor solution for the infiltration process was prepared by mixing the corresponding nitrates and molybdate into deionised water in a concentration of 0.6 mol L^−1^, and then adding citric acid and ethanol to ensure that the solution was able to coat the scaffold.

### 2.2. Cell Fabrication

The SDC powder was compacted into disk-shaped pellets and subjected to calcination at a temperature of 1400 °C for 5 h, resulting in the formation of dense electrolyte pellets. The electrolyte pellets were abraded to a thickness of 300 μm. For the cells with symmetrical structures, consisting of electrodes with either a single phase LSFNM or a physical mixture of SDC and LSFNM in a 70:30 ratio (LSFNM-SDCp), the electrode slurries were sprayed symmetrically onto the two faces of each of the SDC pellets, and subsequently heated to 1000 °C for 2 h to produce the SSOFCs. For the LSFNM-SDCi electrode symmetrical cells, SDC and soluble starch (pore-forming agent) were mixed in isopropanol to form a suspension, which was applied onto both surfaces of the SDC pellet. The porous SDC scaffold was then created by subjecting the pellet to calcination at a temperature of 1250 °C for 5 h. Subsequently, the precursor solution was repeatedly introduced into the porous SDC scaffold until it attained a total penetration of 20 wt %. Finally, the cells underwent calcination at a temperature of 900 °C for 2 h. The bars of LSFNM used in the conductivity and thermal expansion coefficient (TEC) tests were prepared and calcined at 1300 °C for 5 h.

### 2.3. Characterisation Techniques

The phase structures of the produced samples were analysed using X-ray diffraction (XRD) with Cu-Kα radiation (Bruker, D8 Advance, Karlsruhe, Germany). The TECs of materials were tested from 30 to 1000 °C using a heating rate of 5 °C min^−1^ in a normal atmosphere using a Netsch DIL 402C/3/G device. The microstructures of the materials and fuel cells were observed by scanning electron microscopy (CIQTEK, SEM4000, Beijing, China) and high-resolution transmission electron microscopy (HR-TEM, Thermo Fisher, TF-G20, Waltham, MA, USA). The valence states and relative concentrations of the elements on the surfaces of LSFNM particles, both before and after reduction, were analysed by X-ray photoelectron spectroscopy (XPS, Thermo Fisher, Nexsa, Waltham, MA, USA). Using a chemisorption analyser instrument (Builder, PCA-1200, Beijing, China) equipped with a thermal conductivity detector (TCD) to conduct the temperature-programmed reduction (TPR) analysis. The reduction process was initiated by exposing the sample to a mixed gas (10% vol H_2_ in Ar, 30 mL min^−1^) and gradually heating it from 50 °C to 1000 °C at a rate of 10 °C min^−1^.

The area-specific resistance (ASR) of each cell was evaluated by obtaining electrochemical impedance spectra (EIS) using a Princeton impedance spectrum analyser with a steady AC signal amplitude of 10 mV. The EIS measurements were performed across a frequency range spanning from 100 kHz to 0.1 Hz. The electrical conductivities of the bar-shaped samples of LSFNM, calcined at 1300 °C for 5 h, were measured via a four-probe direct current conductivity method using a Keithley 2440 sourcemeter. I-V-P curves of single cells were measured, also using a Keithley 2440 sourcemeter. Hydrogen (or ammonia) at an average flow rate of 50 mL min^−1^ was supplied to the anode side, and the cathode side was exposed to the air.

## 3. Results and Discussion

### 3.1. Basic Characterisation of LSFNM

The LSFN and LSFNM before reduction, and after (r-LSFN and r-LSFNM), under 3% H_2_O-humidified H_2_ at 800 °C for 10 h, were first analysed by XRD in order to determine their phase composition. As shown in Figure 1a, both LSFN and LSFNM exhibited pure cubic perovskite structure prior to reduction, without the formation of other phases, which is consistent with the reported results [26,29,30]. Upon reduction, for the LSFNM, the primary phase remained the cubic perovskite, with secondary phases also being present, including the FeNi_3_ alloy and a minor quantity of the SrLaFeO_4_ oxide phase, whereas the LSFN had decomposed into SrLaFeO_4_ and FeNi_3_ alloy in significant quantities, with the main phase transitioning from LaFeO_3_ to SrLaFeO_4_. This suggests that the incorporation of Mo^6+^ doping significantly enhanced the reduction stability of LSFN and suppressed decomposition under reducing conditions. Figure 1b shows the zoomed-in perspective of four materials within the interval of 30–35°. The peak position of LSFNM exhibits a slight shift towards a smaller angle in comparison to that of LSFN. This phenomenon occurred as a result of the introduction of high-valence Mo into the lattice, which causes the reduction of some Fe species from high-valent to lower-valent states, resulting in lattice expansion. Similarly, in both materials, the main peaks shifted towards smaller angles after reduction, due to the reduction in the state of oxidation on the B-site ions.

To investigate the microstructural features and crystal structure of r-LSFNM, HR-TEM analysis was conducted. Figure 2a shows the TEM images of the r-LSFNM nanoparticles. The average size of the r-LSFNM nanoparticles is approximately 300 nm, and numerous nanoparticles on the exterior of r-LSFNM have an average particle size of ~20 nm. Furthermore, three phases in r-LSFNM were further validated by HR-TEM (Figure 2b,c). The distances between the lattice points in LaFeO_3_, SrLaFeO_4_ and FeNi_3_ alloy were around 0.225, 0.324 and 0.208 nm, in line with (111), (004) and (111) diffraction planes of pertinent phase structures, respectively. Scanning TEM-energy-dispersive X-ray spectroscopy (STEM-EDX) was performed to confirm the distribution of each element of the r-LSFNM particles and to determine the presence or absence of FeNi_3_ nano-alloy exsolution. The corresponding EDX mapping images (Figure 2d) show that the La, Sr, Fe, Ni, Mo and O elements are all dispersed uniformly in the r-LSFNM sample. Furthermore, the partial aggregation of Fe and Ni elements provides evidence for the exsolution of FeNi_3_ alloy particles.

The aforementioned investigations demonstrated the effective enhancement of reduction stability through Mo doping. To explain this phenomenon, XPS analyses were conducted on LSFNM and r-LSFNM. For comparison, XPS spectra of LSFN and r-LSFN are also provided. The XPS data was processed using Thermo Avantage 5.9922 software and fitted using Shirley’s background subtraction approach. Figure 3a shows the XPS spectra of Fe 2p on LSFNM and r-LSFNM. Before reduction, the peaks observed at 709.1 eV and 722.1 eV correspond to Fe^2+^, while the peaks at 710.3 eV and 723.7 eV correspond to Fe^3+^. The peaks at 713.0 eV and 726.3 eV are associated with Fe^4+^, and the peak at 718.5 eV is attributed to a satellite state. In the sample, the ratios of Fe^2+^, Fe^3+^ and Fe^4+^ are 14%, 49% and 37%, respectively. For the reduced LSFNM, an additional peak corresponding to Fe^0^ was observed at 706.4 eV, but the region of the peak that corresponds to metallic iron is relatively small in comparison to those attributed to Fe^2+^, Fe^3+^ and Fe^4+^, demonstrating that only a limited number of ionic forms of Fe were converted into metallic form through reduction. In this sample, the ratios of Fe^2+^, Fe^3+^, Fe^4+^ and Fe^0^ are correspondingly 27%, 46%, 26% and 1%. It was also observed that a portion of Fe^4+^ and Fe^3+^ were reduced to Fe^3+^, Fe^2+^ and Fe^0^. The presence of Fe^0^ indicates the occurrence of metallic iron exsolution from the lattice; however, its proportion is less than 1%, suggesting the exceptional reduction stability of the material. Figure 3b shows the XPS spectra of Fe 2p on LSFN and r-LSFN. The XPS spectrum of LSFN is similar to that of LSFNM. The difference is that the average valence state of Fe cations in LSFN is higher than that of LSFNM due to the Mo doping. For the two reduced samples, obviously, the Fe^0^ and Fe^2+^ contents of r-LSFN were larger than that of r-LSFNM, indicating that Mo doping did effectively improve the reduction stability of the material.

The LSFNM Mo 3d spectra, seen in Figure 3c, exhibit the characteristic Mo 3d_3/2_ and Mo 3d_5/2_ excitations. It is well-established that Mo cations in perovskite oxides tend to adopt a Mo^6+^ oxidation state under oxidising conditions. In the spectra of the LSFNM sample, a broad peak is observed for the Mo 3d_5/2_ (232.5 eV for Mo^6+^) excited states. However, two distinct peaks were observed that corresponded to the Mo 3d_5/2_ energy level (232.6 eV for Mo^6+^ and 231.7 eV for Mo^5+^ excited states) for the r-LSFNM sample. Notably, negligible quantities of Mo^5+^ were detected in the reduced LSFNM. The detection of minor quantities of Mo^5+^ in r-LSFNM indicates that a small fraction of Mo^6+^ cations underwent reduction to Mo^5+^ cations. Through the research of Goodenough [31], we gain valuable insights into various aspects of the study in regard to the overlapping of the Mo^6+^ and Mo^5+^ redox band with the Fe^3+^ and Fe^2+^ couple. Therefore, even in an atmosphere with high reducing agents, it is not practical to completely convert all the Fe^3+^ to Fe^2+^ or Fe^0^. In a reducing environment, it is anticipated that the Mo^6+^ and Mo^5+^ redox couple will maintain the mixed-valent state. This potentially explains the exceptional reduction stability observed in LSFNM.

Adequate electrical conductivity is a prerequisite for electrode materials to be useful for fuel cells. Figure 4a shows the electrical conductivities of the LSFNM as temperature varies in the air. The electrical conductivity of this substance is directly proportional to the temperature, meaning that, as the temperature rises, the conductivity also increases. The LSFNM’s conductivity surpasses 85 S cm^−1^ in the same temperature range, which fully satisfies the needs of an SOFC cathode material [32]. In order to comprehend the reduction processes from LSFNM (or LSFN) to r-LSFNM (or r-LSFN) in the presence of a reducing atmosphere, H_2_-TPR tests were conducted using a carrier gas consisting of 10% H_2_/Ar. Figure 4b displays the relevant data. Similar to the study conducted by Xu et al. [33], three distinct peaks can be observed in the TPR curves for both samples: Peak I at 140–380 °C, Peak II at 390–600 °C and Peak III above 700 °C. The most prominent reduction signals observed in Peak I can be attributed to the reduction of Fe^4+^/Fe^3+^ cations to their lower valences, specifically the transitions of Fe^4+^ to Fe^3+^ and Fe^3+^ to Fe^2+^. This is consistent with the XPS analysis, which showed a decrease in Fe^4+^ and Fe^3+^, and an increase in Fe^2+^ content for the two samples after reduction, especially for the LSFN. According to the report by Kim et al. [34], it has been established that the process of converting Ni^3+^ to Ni^2+^ occurs at ~450 °C, followed by the Ni^2+^ turning to metallic nickel at ~600 °C. Furthermore, for LSFNM, it has been documented that a decrease in Mo^6+^ occurs at approximately 596 °C [35]. Given this information, it is logical to assume that Peak II is linked to the processes of reduction of Ni^3+^ to Ni^2+^, Ni^2+^ to Ni^0^ and an additional reduction process of Mo^6+^ to Mo^5+^ for the LSFNM. The presence of Mo^5+^ can also be demonstrated by the XPS analysis in Figure 3c. Owing to the low concentrations of Mo and Ni in LSFN and LSFNM, the intensity levels of Peak II appear relatively weaker. For Peak III, the readings above 700 °C in the TPR curves may be equivalent to the reduction of Fe^2+^ to Fe^0^, suggesting that Fe and Ni have been dissolved from the LSFN and LSFNM, which is consistent with the STEM-EDX mapping results described above [36]. It is worth mentioning that according to the XPS analysis, r-LSFNM and r-LSFN had low Fe^0^ contents of 1% and 9%, respectively, but it was found from the TPR curves that both samples consumed a large amount of H_2_, indicating that abundant Fe^0^ should be generated. This is because the reduced samples used for XPS analysis were treated at 800 °C, while the reduction process from Fe^2+^ to Fe^0^ (Peak III) began to rise rapidly at ~900 °C or even higher. By comparing the TPR processes of LSFN and LSFNM, we find that the hydrogen consumption (peak area) of LSFN is much higher than that of LSFNM, regardless of which reduction process (Peak I, Peak II or Peak III), implying that more Fe cations were reduced in LSFN, which is consistent with the XPS results. The onset temperature of Peak I of LSFNM is lower than that of LSFN, demonstrating that Mo doping can promote the reduction of Fe cations to lower valence states. Looking at Peak III, the onset temperature of LSFN is lower than that of LSFNM. Moreover, LSFN peaks at ~920 °C, while LSFNM does not peak even at 1000 °C. This suggests that the Mo doping makes the exsolution of metallic Fe relatively difficult, thus ensuring the reduction stability of LSFNM.

### 3.2. Basic Characterisation of LSFNM-Infiltrated SDC

It has been widely demonstrated that perovskite materials that have been infiltrated into porous electrolyte material scaffolds to form composites can provide an enhanced pathway for ion transfer, making them a more advantageous choice for SSOFC electrodes [17,37]. Therefore, we used LSFNM-SDCi as the electrode for the DA-SSOFC in this work. Firstly, it was necessary to understand some of the basic properties of the LSFNM-SDCi, so we assessed the phase reaction between the two components (LSFNM and SDC). The XRD patterns of LSFNM, SDC and LSFNM-SDCi (calcined at 900 °C in air) are presented in Figure 5a. For the LSFNM-SDCi, the diffraction peaks can be precisely indexed by considering a combination of LSFNM and SDC. The extra small peak at 27° is a Kβ peak [38]. This observation suggests that there was no chemical interaction between the LSFNM and SDC after calcination at 900 °C. 

During the operation of an SOFC, the existence of a discrepancy in the TEC between the electrolyte and electrodes can lead to substantial stress and deformation, resulting in a significant decrease in performance and stability. To ensure the consistent functioning of the SOFCs at high temperatures, it is crucial to establish good thermomechanical compatibility between the electrode and electrolyte. Consequently, the thermal expansion curves of LSFNM and LSFNM-SDCi were measured and are shown in Figure 5b. The average TEC of pure LSFNM in the temperature range of 100–950 °C was 13.5982 × 10^−6^ K^−1^ based on the d(ΔL/L_0_) temperature curve, but for LSFNM-SDCi, the value was 12.7386 × 10^−6^ K^−1^. Both of these materials exhibit a TEC that is matched to the electrolyte SDC. In particular, the TEC of LSFNM-SDCi is highly similar to that of SDC (a TEC of 12.6 × 10^−6^ K^−1^) [39], so the electrolyte can be better matched to the electrode, ensuring the long-term operation of the SSOFC. 

To visually demonstrate that LSFNM is well impregnated into the SDC backbone, SEM characterisation was conducted to examine the minuscule structures of the samples. As shown in Figure 5c,d, the SDC scaffold consisted of grains with a size ranging from 200–500 nm, exhibiting polygonal shapes, with the particles being effectively fused together to create a seamless structure within the porous scaffold. Following the infiltration of LSFNM into the porous SDC scaffold, and subsequent heating at 900 °C, the inner walls of the scaffold exhibited a dense distribution of LSFNM nanoparticles. A substantial amount of LSFNM nanoparticles adhered to a continuous SDC scaffold, effectively increasing the reaction sites and thereby enhancing the ORR activity and the HOR activity for the cathode and anode, consequently improving the cell’s performance.

### 3.3. Electrocatalytic Performance

We first evaluated the ORR activity of LSFNM using EIS. For better comparison, pure LSFNM as well as physically mixed LSFNM and SDC (LSFNM-SDCp) were both tested. The ASR, obtained through EIS, served as the primary parameter for evaluating the ORR activity of the materials. The ASRs of the SDC symmetrical cell with the LSFNM-series electrode were measured under an air atmosphere within the range of 550–750 °C, based on a symmetrical configuration. Figure 6a,c show Nyquist plots of symmetrical cells utilising various electrode materials at 700 °C in air. All EIS data were fitted using a R_ohm_ − (R_E1_ − CPE_1_) − (R_E2_ − CPE_2_) equivalent circuit. It can be observed that the fitted values closely match the actual measured results, indicating a good agreement between them. Typically, the impedance response in the Nyquist plots exhibits distinct semicircles at different frequencies, which can be attributed to various processes. These processes include gas diffusion and surface adsorption, which occur at lower frequencies, surface diffusion and oxygen dissociation at intermediate frequencies and the charge transfer process at higher frequencies. Figure 6d illustrates the variation in ASR values for different electrode materials at different temperatures. When utilising LSFNM as the electrode, the ASRs obtained at temperatures of 550, 600, 650, 700 and 750 °C were found to be 52.3, 13.8, 4.1, 1.3 and 0.48 Ω cm^2^, respectively. On the other hand, the LSFNM-SDCp composite electrode exhibited ASRs of 22.4, 6.2, 2.1, 0.74 and 0.29 Ω cm^2^ at the same respective temperatures. These results indicate a moderate improvement in ASRs compared to those of the LSFNM electrode, which can be ascribed to the improved oxygen-ion conductivity resulting from the incorporation of SDC into the electrode structure. This, in turn, leads to an expansion of the reaction sites for ORRs and an increase in electrochemically active sites. For LSFNM-SDCi, the effective surface modification of SDC with nano-LSFNM enhances oxygen dissociation and surface diffusion capabilities. As a result, ASRs of 0.94, 0.40, 0.18, 0.09 and 0.04 Ω cm^2^ at the same respective temperatures were obtained, providing compelling evidence of its superior ORR activity. Figure 6d also illustrates the Arrhenius curves of the ASRs for the symmetrical cell with different materials at the temperatures of 550–750 °C. The activation energy (Ea) values for the symmetrical cell with the LSFNM, LSFNM-SDCp and LSFNM-SDCi electrodes are 164, 151 and 103 kJ mol^−1^, respectively. Those Ea values imply the temperature dependence of electrode performance. LSFNM-SDCi exhibits the lowest Ea value, indicating that its performance degradation is minimal at low temperatures.

To evaluate the performance of LSFNM-SDCi electrodes in a real SOFC, SDC electrolyte-supported SOFCs with LSFNM-SDCi symmetrical electrodes were fabricated and tested, firstly using H_2_ fuel. As shown in Figure 7a, PPDs of 618, 494, 388, 293 and 218 mW cm^2^ were achieved over the range of 800–600 °C. The excellent power outputs obtained demonstrate that the LSFNM-SDCi symmetrical electrode is promising. The corresponding impedance spectra of the SSOFC under open-circuit voltage (OCV) conditions were also measured and are shown in Figure 7b. In general, the point where the semicircle intersects with the horizontal axis at higher frequencies reflects the ohmic resistance (R_o_) of the cell, whereas the intersection at lower frequencies represents the total electrode resistance (R_t_). The difference between R_t_ and Ro corresponds to the polarisation resistance (R_p_) of the SOFC electrode, where Ro is primarily composed of the electrolyte’s ohmic resistance and the contact resistance between the electrode and electrolyte, with the former being predominant. R_p_ is mainly associated with concentration polarisation and activation polarisation during the electrode catalytic reaction. As depicted in Figure 7b, the R_p_ values are 0.029, 0.045, 0.067, 0.102 and 0.163 Ω cm^2^ at temperatures of 800, 750, 700, 650 and 600 °C, respectively.

To further prove the superiority of impregnated LSFNM, SSOFCs with both pure LSFNM and LSFNM-SDCp as electrodes were also tested and compared. Figure 7c presents the I-V and I-P curves of the SSOFCs with different electrodes measured in H_2_. The SSOFCs with LSFNM, LSFNM-SDCp and LSFNM-SDCi electrodes attained PPDs of 435, 466 and 618 mW cm^2^ at 800 °C, respectively. Obviously, the SSOFC with the LSFNM-SDCi electrode performed the best. Combined with the results of EIS tested in air (Figure 6), it can be seen that the LSFNM-SDCi electrode is not only an excellent ORR catalyst, but also exhibits good activity for hydrogen oxidation reactions (HORs), further demonstrating that it is an excellent SSOFC electrode material. To better compare the PPDs of various SSOFCs at different temperatures, Figure 7d lists the PPDs of all SSOFCs at different temperatures. It is evident that the introduction of the SDC phase effectively enhances the performance of the fuel cell. Additionally, the electrode fabricated by the infiltration method exhibits better performance than that prepared by the physical mixing method. This can be attributed to the fact that the continuous SDC scaffold provides a rapid pathway for oxygen ion diffusion, while the infiltrated LSFNM nanoparticles attached to the SDC scaffold improve the efficiency of separating oxygen molecules on the surface and the movement of molecules over the surface. Furthermore, smaller electrode particles have been reported to be advantageous for the ORR process [40,41]. So, as expected, LSFNM-SDCi delivered an excellent electrochemical performance when used as an electrode for SSOFC.

It has been reported that metallic Ni and Fe are good catalysts for NDR and HOR [42,43]. By applying ammonia as a fuel, the reaction that occurs on the anode side can be summarised into three key steps: (1) NH_3_ adsorbs onto the anode, (2) adsorbed NH_3_ decomposition is facilitated by catalytically active metal particles, (3) an electrochemical reaction between H_2_ and O^2−^ [44] occurs. Based on these steps, we can see that it is crucial for the anode catalyst to exhibit outstanding electrocatalytic activity for both HOR and NDR. Hence, the catalytic activity assessment of r-LSFNM for NDR was conducted in a mixture of 10% NH_3_-Ar, and the outcomes are depicted in Figure 8a. The r-LSFNM catalyst exhibited a near 100% rate of NH_3_ conversion at temperatures ranging from 750–800 °C. The conversion rate decreased at temperatures below 700 °C, with rates of 93%, 85%, 56% and 32% observed at 700, 650, 600 and 550 °C, respectively. The high NH_3_ conversion rate can be attributed to the exsolved FeNi_3_ alloy nanoparticles, which are excellent catalysts for NDR and can accelerate the catalytic decomposition of ammonia. Additionally, Ni has been widely recognised as one of the most efficient catalytic metals for HOR [45]. As a result, LSFNM-SDCi exhibits substantial electrocatalytic activity for both NDR and HOR. These results guarantee satisfactory performance of the SOFC when using ammonia fuel.

Figure 8b shows the I-V and I-P curves of an SSOFC with an LSFNM-SDCi electrode operating with NH_3_ fuel. PPDs of 487, 360, 225, 125 and 56 mW cm^2^ were achieved at 800, 750, 700, 650 and 600 °C, respectively. Although the PPDs obtained are lower than those achieved when using H_2_ fuel, the power outputs of the SSOFC with the LSFNM-SDCi electrode using NH_3_ fuel are of higher quality than the majority of the materials that are currently accessible (Table 1). Figure 8c illustrates the corresponding impedance values of the SSOFC with an LSFNM-SDCi electrode under OCV conditions. The impedance spectra are highly similar to those observed with H_2_ fuel, but the R_p_ values are higher, especially at low operating temperatures. It is supposed that this is because NH_3_ cannot be directly involved in the SOFC power generation process, and the additional decomposition process increases R_p_ values. Figure 8d shows the PPDs of all SSOFCs at different temperatures. Consistent with the findings obtained when hydrogen was employed as the fuel, the SSOFC with the LSFNM-SDCi electrode demonstrated the most favourable performance. The PPDs of SSOFCs with LSFNM, LSFNM-SDCp and LSFNM-SDCi are 335, 363 and 487 mW cm^−2^ at 800 °C, respectively. It can be concluded that the SSOFC showed lower PPDs when operating with NH_3_ fuel, as compared with H_2_ fuel, because the generation of N_2_ from NH_3_ decomposition decreases the concentration of H_2_. Compared to the PPDs of the SSOFCs using H_2_ fuel, the percentage degradation ratios of PPDs of the SSOFCs with LSFNM, LSFNM-SDCp and LSFNM-SDCi electrodes fuelled by NH_3_ were 23.0%, 22.1% and 21.2%, respectively, which are all lower than that reported using a Ni-SDC anode (29%) [17]. This further demonstrates the superiority of LSFNM-SDCi as the anode material for DA-SSOFCs.

In addition to the power output of a fuel cell, long-term stability is another important indicator of SOFC performance. Therefore, the stability of the SSOFC with the LSFNM-SDCi electrode in H_2_ and NH_3_ fuels was investigated. Figure 9a presents the variations in cell voltage with operating time (at 700 °C) at both a constant current of 300 mA cm^−2^ fuelled by H_2_ and a constant current of 100 mA cm^−2^ fuelled by NH_3_. The SSOFC first ran on H_2_ fuel for 150 h, then switched to NH_3_ fuel for an additional 200 h, then finally switched back to H_2_ for another 50 h. The SSOFC demonstrated robust stability throughout the 400 h of operation. The voltage fluctuations observed during the use of NH_3_ fuel can be attributed to the unstable flow rate of NH_3_. The SSOFC exhibited a degradation rate of less than 0.0003 V h^−1^ throughout the entire 400 h of operation, with a voltage decay rate of 0.0004 V h^−1^ observed during the 200 h while operating with NH_3_ as the fuel. Furthermore, for comparative analysis, the stability of SSOFC with LSFN-SDCi was also evaluated, revealing significantly lower stability when compared to the SSOFC with the LSFNM-SDCi electrode. Specifically, a voltage decay rate of 0.011 V h^−1^ was observed during the 50 h operational period using NH_3_ fuel. This may be due to the relatively poor phase stability of the LSFN anode after the reduction, leading to its substantial decomposition into the SrLaFeO_4_ phase under a reducing atmosphere (Figure 1). Notably, SrLaFeO_4_ exhibits a higher TEC of 14.3 × 10^−6^ K^−1^ compared to LSFN and SDC [48,49], resulting in mechanical and structural degradation of the anode, which further indicates the superior durability of the Mo-doped LSFNM electrode. After the long-term stability test, the microscopic morphology of the single cell was observed by SEM. As shown in Figure 9b,c, both the anode and cathode structures exhibit high porosity and demonstrate excellent contact with the electrolyte, thus confirming their favourable chemical compatibility with the electrolyte.

## 4. Conclusions

In conclusion, the LSFNM-based electrode ensures power generation efficiency while significantly improving the stability of the DA-SSOFC. Mo doping largely enhances the reduction stability of the material. The phase structure of LSFN was severely damaged after reduction at 800 °C. However, after treatment in a reduced atmosphere, the main cubic perovskite phase of LSFNM was retained, accompanied by the separation of some FeNi_3_ nanoparticles that were attached to the surface of the substance, which greatly improved the electrocatalytic activity of LSFNM on the HOR and NDR. In addition, the LSFNM-infiltrated SDC electrode exhibited high ORR activity with an ASR of 0.04 Ω cm^2^ at 750 °C. Therefore, the single cell with the LSFNM-infiltrated SDC symmetrical electrode shows inspiring performance, with a PPD of 487 mW cm^−2^ at 800 °C operating with NH_3_ fuel. Most significantly, this kind of DA-SSOFC exhibited superb durability without obvious performance degradation for over 400 h at 700 °C. This work introduces a straightforward and efficient electrode material for DA-SSOFCs, which may speed up their commercialisation in the application of DA-SSOFC technology.

## Figures and Tables

**Figure 1 nanomaterials-14-00673-f001:**
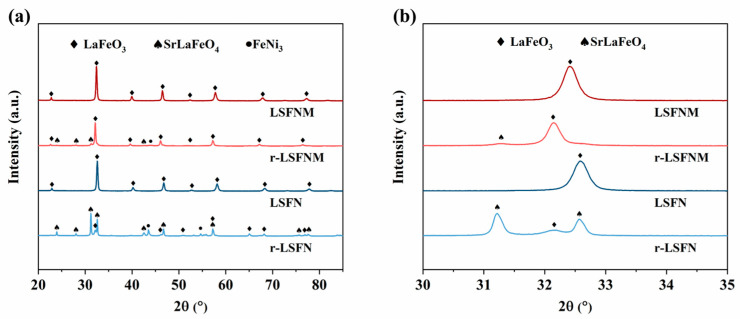
(**a**) XRD characterisation of LSFNM, r-LSFNM, LSFN and r-LSFN, and (**b**) the magnified view in the range of 30–35°.

**Figure 2 nanomaterials-14-00673-f002:**
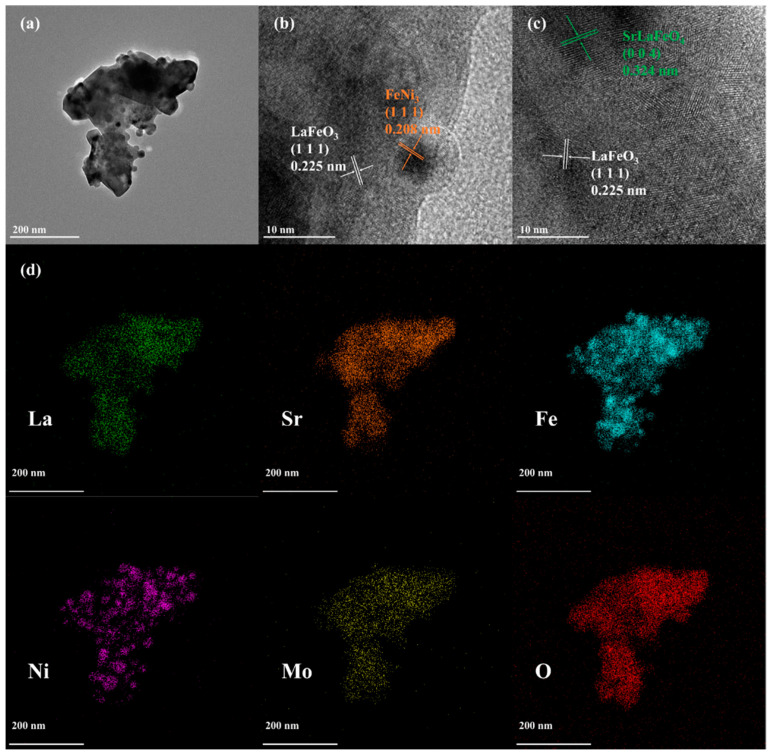
(**a**) TEM image, (**b**,**c**) HR-TEM images and (**d**) STEM-EDX mapping of r-LSFNM sample.

**Figure 3 nanomaterials-14-00673-f003:**
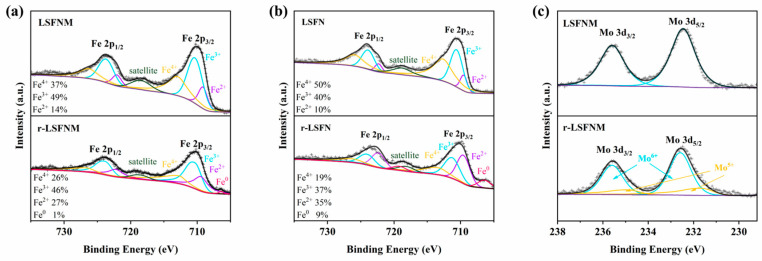
XPS spectra of Fe 2p on (**a**) LSFNM and r-LSFNM, (**b**) LSFN and r-LSFN, and (**c**) Mo 3d on LSFNM and r-LSFNM.

**Figure 4 nanomaterials-14-00673-f004:**
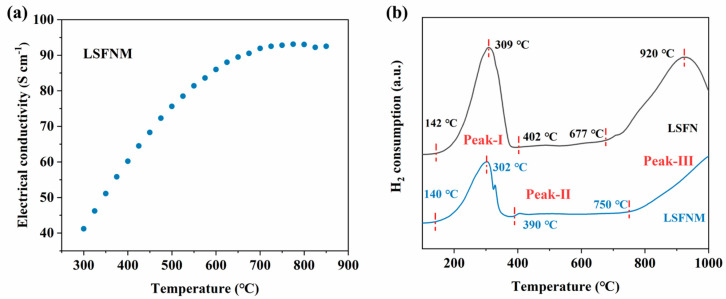
(**a**) Electrical conductivities of LSFNM and (**b**) H_2_-TPR curves of LSFN and LSFNM.

**Figure 5 nanomaterials-14-00673-f005:**
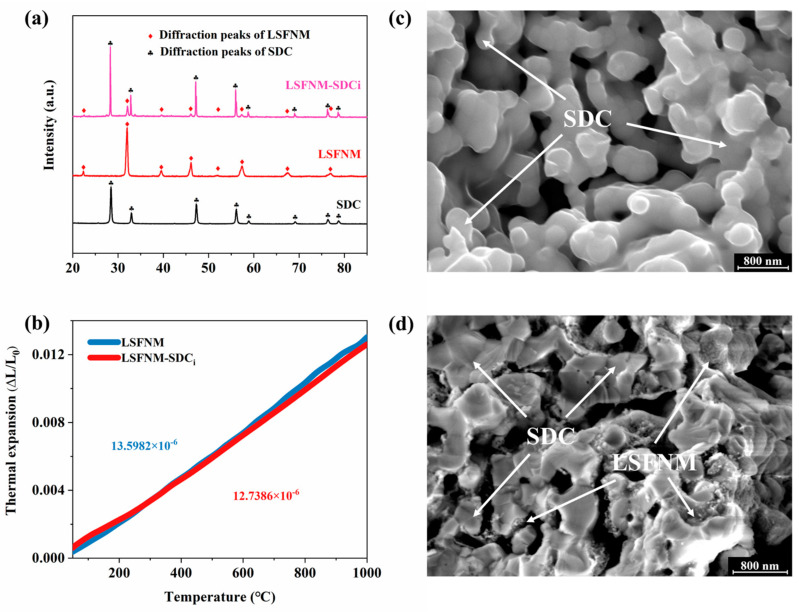
(**a**) XRD patterns of LSFNM, SDC and LSFNM-SDCi. (**b**) TEC curves of LSFNM and LSFNM-SDCi. SEM images of (**c**) porous SDC scaffold and (**d**) LSFNM-SDCi.

**Figure 6 nanomaterials-14-00673-f006:**
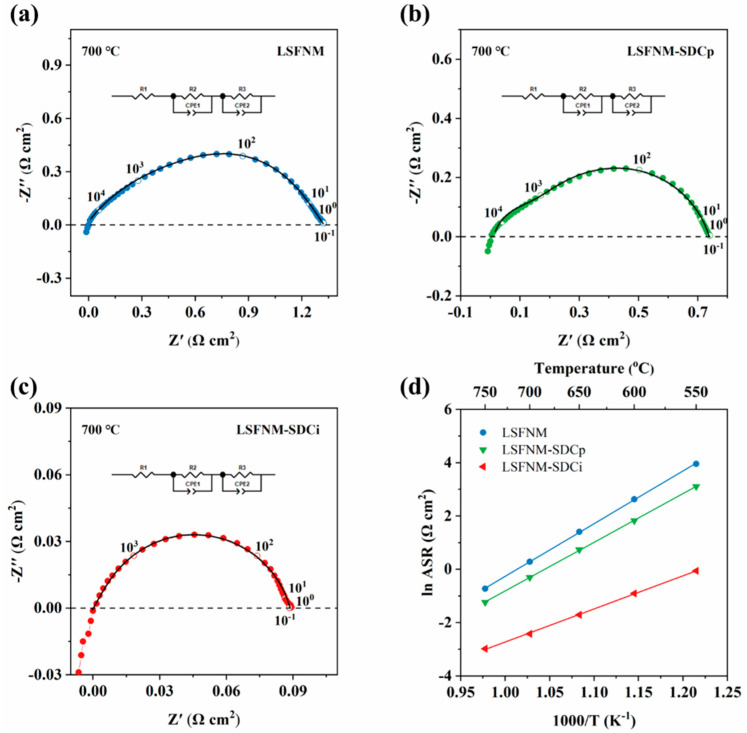
The Nyquist plots for the SDC electrolyte symmetrical cells with (**a**) LSFNM, (**b**) LSFNM-SDCp and (**c**) LSFNM-SDCi electrodes measured at 700 °C in air. (**d**) Arrhenius plots of ASRs for all three electrodes at the temperatures of 550–750 °C.

**Figure 7 nanomaterials-14-00673-f007:**
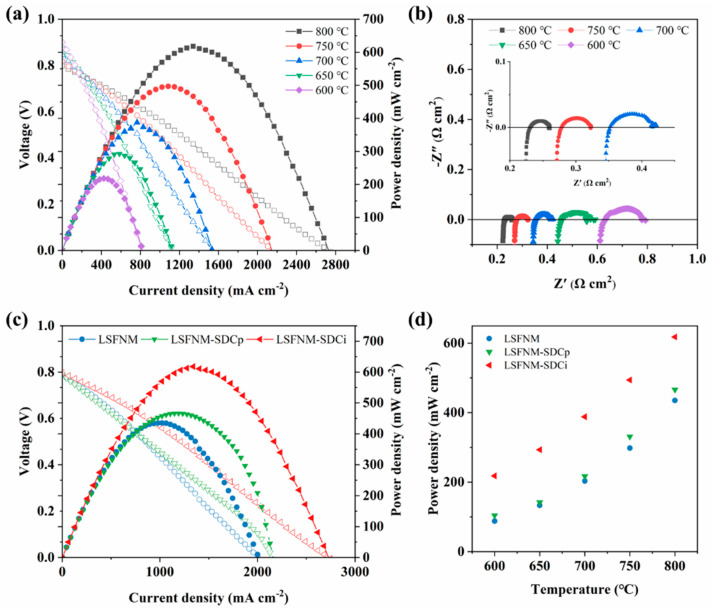
(**a**) I-V-P curves and (**b**) corresponding EIS values of the single cell (LSFNM-SDCi|SDC|LSFNM-SDCi) operating with H_2_. (**c**) I-V-P curves of the SSOFCs with the different electrodes operating with H_2_ at 800 °C. (**d**) PPDs of SSOFCs with different electrode materials at 600–800 °C.

**Figure 8 nanomaterials-14-00673-f008:**
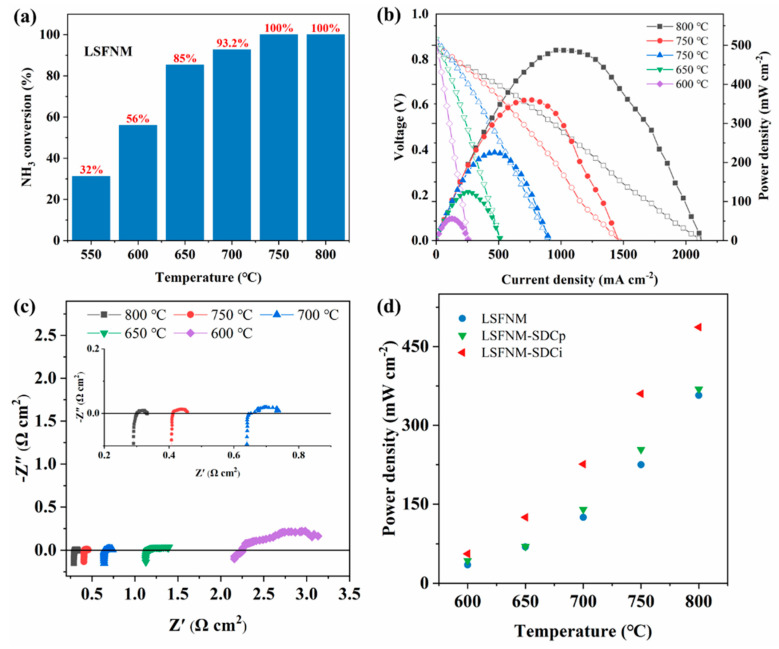
(**a**) The catalytic activity of r-LSFNM for NDR at 550–800 °C. (**b**) I-V-P curves and (**c**) EIS curves under OCV conditions of the SSOFC (LSFNM-SDCi|SDC|LSFNM-SDCi) operating with NH_3_. (**d**) PPDs of SSOFCs with different electrodes at 600–800 °C.

**Figure 9 nanomaterials-14-00673-f009:**
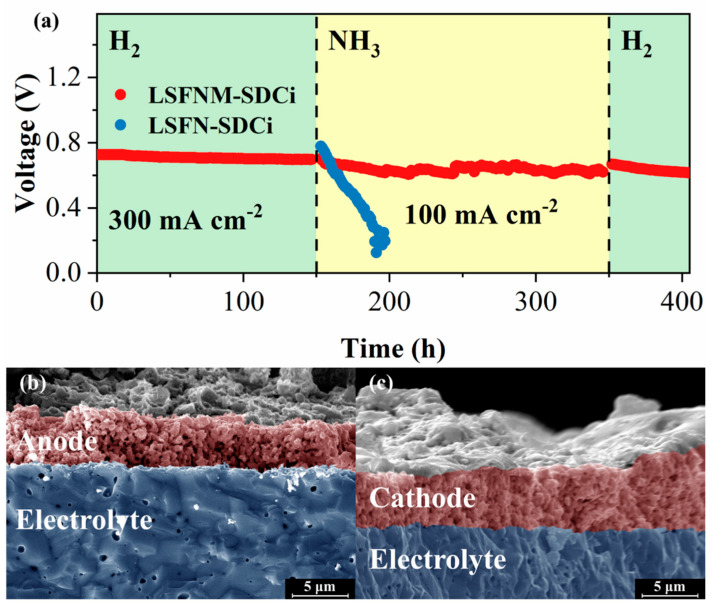
(**a**) Stability tests of the SSOFCs with LSFNM-SDCi and LSFN-SDCi electrodes at 700 °C. (**b**,**c**) SEM images of cross-sections of the SSOFC with LSFNM-SDCi electrode, after the stability test.

**Table 1 nanomaterials-14-00673-t001:** The performance comparison of DA-SOFCs.

Electrolyte	Anode	Cathode	Electrolyte Thickness (µm)	PPD at 800 °C (mW cm^−2^)	Ref.
SDC	LSFNM-SDCi	LSFNM-SDCi	300	487	This work
SDC	Pr_0.6_Sr_0.4_Co_0.2_Fe_0.78_O_3−δ_	BaCo_0.4_Fe_0.4_Zr_0.1_Y_0.1_O_3−δ_	400	288	[15]
SDC	Pr_0.6_Sr_0.4_Co_0.2_Fe_0.75_Ru_0.05_O_3−δ_	BaCo_0.4_Fe_0.4_Zr_0.1_Y_0.1_O_3−δ_	400	374	[15]
SDC	La_0.52_Sr_0.28_Ti_0.94_Ni_0.06_O_3−δ_-SDC	Ba_0.5_Sr_0.5_Co_0.8_Fe_0.2_O_3−δ_	350	161	[17]
SDC	La_0.52_Sr_0.28_Ti_0.94_Co_0.06_O_3−δ_-SDC	Ba_0.5_Sr_0.5_Co_0.8_Fe_0.2_O_3−δ_	350	98	[17]
SDC	La_0.52_Sr_0.28_Ti_0.94_Ni_0.03_Co_0.03_O_3−δ_-SDC	Ba_0.5_Sr_0.5_Co_0.8_Fe_0.2_O_3−δ_	350	361	[17]
La_0.9_Sr_0.1_Ga_0.8_Mg_0.2_O_2.85_	Ni-SDC	Sm_0.5_Sr_0.5_CoO_3−δ_	500	118	[46]
La_0.9_Sr_0.1_Ga_0.8_Mg_0.2_O_2.85_	Ni(40)Fe(60)-SDC	Sm_0.5_Sr_0.5_CoO_3−δ_	500	250	[46]
6mol% YSZ	Ni-YSZ	Ag	400	75	[47]

## Data Availability

The data presented in this study are available on request from the corresponding authors.

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
