# Peer review of "A High-Performance and Durable Direct-Ammonia Symmetrical Solid Oxide Fuel Cell with Nano La_0.6_Sr_0.4_Fe_0.7_Ni_0.2_Mo_0.1_O_3−δ_-Decorated Doped Ceria Electrode"

_nanomaterials, 2024, doi:10.3390/nano14080673_

Round 1

Reviewer 1 Report

Comments and Suggestions for Authors

This manuscript, entitled "A High-Performance and Durable Direct-Ammonia Symmetrical Solid Oxide Fuel Cell with Nano La0.6Sr0.4Fe0.7Ni0.2Mo0.1O3-δ Decorated Doped Ceria Electrode", is well written and informative. It presents interesting and important findings on the use of Mo-doped LSFNM materials as a symmetrical electrode for DA-SOFCs for both NH3 decomposition and H2 oxidation reactions.

I recommend it for publication in “nanomaterials”, with the following minor corrections and after the correction of some typographic errors:

·        - Abstract: line 15. Please, change “Nh3” to “NH3” (also on line 47).

·       -  Line 41: Please, change “its Storage ….” to “its storage…”

    - Line 80: Change “a Mo-doped La0.6Sr0.4Fe0.8Ni0.2O3-δ (LSFNM) catalyst” to “a Mo-doped La0.6Sr0.4Fe0.8Ni0.2O3-δ (La0.6Sr0.4Fe0.7Ni0.2Mo0.1O3-δ, LSFNM) catalyst”

·  - Line 115: Change “calcination at a temperature of 1250 °C 5 hours” to “subjecting it to calcination at a temperature of 1250 °C for 5 hours” or “calcination at 1250 °C for 5 hours”.

·    - Line 123: Check the expression “filtered through a D8 Advance..”, in my opinion it is more correct to use "using a D8 Advance Bruker diffractometer".

·        -  Please, define in the text the acronyms used in the lines 123 “TEC”, line 131 “TCD” and line “ASR”, because it is confusing for the reader.

·         - Line 145: “Basic Properties of LSFNM”, in my opinion is more correct “Basic Characterization of LSFNM”.

·        -  Lines 154 and 172: Change “LaFeO3” to “LaFeO3

·        -  In my opinion, in Figure 1b the authors should include the identification of the peaks so as not to confuse the reader. They could include in the Figure 1b the same symbols used in Figure 1a.

·        -  Line 173: Please, review the sentence, since according to the Figure 2b-c, reflections 111, 004 and 111 correspond to the LaFeO3, FeNi3 and SrLaFeO4 phases.

·         - Check the following typographical errors inline 187: change “Fe2p” and Mo5d” to “Fe 2p” and “Mo 5d”. Line 188: Change “Fe2+” to “Fe2+” and in line 189, change “Fe3+ and Fe4+” to “Fe3+ and Fe4+”. Lines 200-201: change “Mo 3d5/2 and Mo 3d5/2” to “Mo 3d3/2 and Mo 3d5/2”. In line 203 missing a space “232.5eV”.

·        -  Line 228: change “to Fe2” to “to Fe2+”.

·         - Line 235: “For Peak-III,..” left over in the sentence, and in line 237 is mare correct “STEM-EDX mapping”.

·      -    Line 240: “Basic Properties of LSFNM infiltrated SDC”, in my opinion is more correct “Basic Characterization and Properties of LSFNM infiltrated SDC”.

·        -  In figure 5a you can see additional peaks in the area between 25-35º, although they are minority peaks, the authors have not identified them. Do these peaks correspond to other phases present?

·     -     The authors state that LSFNM nanoparticles have a uniform size distribution of 20-30 nm. However, in the figures 5c-d show images with a magnification of 800 nm, it is impossible to determine said particle size with these images. The authors should include other images with higher magnification in the manuscript to be able to affirm this.

·         - Line 332: Please, change “0.163 Ω cm2” to “0.163 Ω cm2

·      -    Lines 366 and 368 change “NH3” to “NH3”.

·       -   Line 397, the reference [21] is correct? or is the reference [48].

·         - Table 1, authors should include units of electrode thickness (µm) and PPD (mW·cm-2) in the table. The PPD value of LSFNM-SDCi cathode in this work not is 418 mW·cm-2 is 487 mW·cm-2

·          - Line 470, in reference [7] change “SrCo0.8Nb0.1Ti0.1O3−δ” to “SrCo0.8Nb0.1Ti0.1O3−δ

·         -  Line 520, in reference [30] change “CO2” to “CO2

Comments on the Quality of English Language

This manuscript, entitled "A High-Performance and Durable Direct-Ammonia Symmetrical Solid Oxide Fuel Cell with Nano La0.6Sr0.4Fe0.7Ni0.2Mo0.1O3-δ Decorated Doped Ceria Electrode", is well written and informative. It presents interesting and important findings on the use of Mo-doped LSFNM materials as a symmetrical electrode for DA-SOFCs for both NH3 decomposition and Hoxidation reactions.

Reviewer 2 Report

Comments and Suggestions for Authors

1. "The LSFNM and LSFN precursor solutions for the infiltration process were prepared by mixing the corresponding nitrates"

- How did you introduce the molybdenum in this solution?

2. "XRD measurement in order to analyze their phase structure and composition"

- What is the difference between the phase structure and the phase composition? Did you analyze the phase structure indeed?

3. "after reduction (r-LSFN and r-LSFNM) under wet H2 at 800 °C"

-What do you mean "wet H2", what amount of water it contained, and how did you obtain this wet H2?

4. "Using a PCA-1200 instrument equipped with a TCD" 

- What kind of instrument is PCA-1200? Please give the full names first for TCD and ASR. 

5."It suggests that the incorporation of Mo6+ doping significantly enhances the reduction stability of LSFN and suppresses decomposition under reducing conditions."

- If so, it is necessary to add the TPV curve of Mo-free LSFN in Fig. 4b.

6. “The aforementioned experiments demonstrated the effective enhancement of reduction stability through Mo doping. To explain this phenomenon, XPS analysis was conducted on LSFNM and r-LSFNM.”

 -         In fact, in the lack of comparison with corresponding data for LSFN the aforementioned TEM data demonstrated the results of LSFNM degradation only. In order to explain this phenomenon, you had to compare the XPS spectra of r-LSFN and r-LSFNM.

7. Fig. 4. A comparison of the amounts of H2 consumed during processes I, II, and III with the amounts of corresponding cations in LSFNM and with XPS data would allow you to interpret the H2-TPR data more precisely.

8. According to Fig. 5, the SDC ceramics are formed by the connected SDC grains. The LSFNM particles in LSFNM-SDC are located at the surface of SDC grains and, hence, should not affect the thermal expansion of the SDC framework. If so, please explain why the TECs of SDC and LSFNM-SDC composite are not the same.

Comments on the Quality of English Language

Only minor revision of English is needed. 

Reviewer 3 Report

Comments and Suggestions for Authors

Dear Authors! Thank you for submitting your manuscript to Nanomaterials. I have read it and I think that the present article contains very interesting results on the development of the active nanostructured electrode based on La0.6Sr0.4Fe0.7Ni0.2Mo0.1O3-δ for the applications in direct ammonia SOFCs. It is certainly within the scope of the journal and would be of interest to the research community. In the meantime, before publication, I suggest the following corrections to the manuscript:

1. Please revise the introduction section by adding the information about the previous investigations of La0.6Sr0.4Fe0.8Ni0.2O3-δ and La0.6Sr0.4Fe0.7Ni0.2Mo0.1O3-δ.

2. Please, shorten the last paragraph of the introduction - it should include the conclusions from the literature part, emphasize the research gap and the aims of the present work. Not the conclusions and results of this paper.

3. Please give the details of the synthesis of La0.6Sr0.4Fe0.7Ni0.2Mo0.1O3-δ powders. How was SDC obtained?

4. The manuscript contains many typing errors. Please check and correct them carefully.

5. Finally, I would like to say that the very poor quality of the English language spoils the overall pleasant impression of the manuscript. Please have the manuscript proofread by a native speaker.

Comments on the Quality of English Language

The text of the manuscript requires extensive proofreading by a native speaker to eliminate grammatical and semantic errors. 

Round 2

Reviewer 3 Report

Comments and Suggestions for Authors

Dear Authors! Thank you for submitting the revised version of your manuscript and for considering my comments! I consider, the manuscript has been properly revised.